# Effect of Electric Field Regulation on Laser Damage of Composite Low-Dispersion Mirrors

**Yuhui Zhang** [1,2,3], **Yanzhi Wang** [1,3,*], **Hongbo He** [1,3,*], **Ruiyi Chen** [1,2,3], **Zhihao Wang** [1,2,3], **Dawei Li** [1,3], **Meiping Zhu** [1,3], **Yuanan Zhao** [1,3], **Yunxia Jin** [1,3], **Kui Yi** [1,3], **Yuchuan Shao** [1,3], **Yuxin Leng** [4,5], **Ruxin Li** [4,5] and **Jianda Shao** [1,3,5,6,*]

1   Laboratory of Thin Film Optics, Shanghai Institute of Optics and Fine Mechanics, Shanghai 201800, China; yhzhang@siom.ac.cn (Y.Z.); chenruiyi@siom.ac.cn (R.C.); darlingwangzh@126.com (Z.W.); lidawei@siom.ac.cn (D.L.); bree@siom.ac.cn (M.Z.); yazhao@siom.ac.cn (Y.Z.); yxjin@siom.ac.cn (Y.J.); kyi@siom.ac.cn (K.Y.); shaoyuchuan@siom.ac.cn (Y.S.)
2   Center of Materials Science and Optoelectronics Engineering, University of Chinese Academy of Sciences, Beijing 100049, China
3   Key Laboratory of Material for High Power Laser, Shanghai Institute of Optics and Fine Mechanics, Shanghai 201800, China
4   State Key Laboratory of High Field Laser Physics, Shanghai Institute of Optics and Fine Mechanics, Shanghai 201800, China; lengyuxin@siom.ac.cn (Y.L.); ruxinli@mail.shcnc.ac.cn (R.L.)
5   CAS Center for Excellence in Ultra-Intense Laser Science, Shanghai 201800, China
6   Hangzhou Institute for Advanced Study, Hangzhou 310024, China
*   Correspondence: yanzhiwang@siom.ac.cn (Y.W.); hbhe@siom.ac.cn (H.H.); jdshao@mail.shcnc.ac.cn (J.S.)

**Abstract:** Low dispersion mirrors are important because of their potential use in petawatt (PW) laser systems. The following two methods are known to increase the laser-induced damage threshold of low dispersion optical components: use of a wide-bandgap-material protective layer and control of electric field distribution. By controlling the electric field distribution of composite low-dispersion mirrors (CLDM), we shift the electric field peaks from the material interface into the wide-bandgap material. However, the damage threshold of modified-electric-field composite low dispersion mirror (E-CLDM) does not increase. Damage morphology shows that the initial damaged layer is $Ta_2O_5$. An immediate cause is the enhancement of the electric field in internal layers caused by surface electric field regulation. Theoretical calculations show that the damage threshold of CLDM or E-CLDM is determined by the competition results of bandgap and the electric field of layer materials. The CLDM with different materials or different protective layer periods can be optimally designed according to the electric field competition effect in the future.

**Keywords:** low dispersion mirror; laser damage; PW laser system

## 1. Introduction

The development of high-power petawatt (PW) laser promotes the development of physics, medicine, biology and materials science [1–5]. However, the generation of high-power pulses is constrained by the laser-induced damage threshold (LIDT) of optical elements. In particular, the LIDT of a low-dispersion mirror (LDM) is a weak point in the development of high-power ultrafast systems [6]. LDMs that can afford a broadband high-reflection (HR) spectrum, high LIDT, and stable group delay dispersion (GDD) play an essential role for PW lasers with pulses in the femtosecond regime [7]. However, reflection bandwidth, dispersion and damage threshold are interdependent, restricting each other [8]. Thereby, the LIDT of broad-bandwidth, low-dispersion mirrors have become a recognized research focus [8–16].

Quarter-wave optical thickness (QWOT) mirrors are the most commonly used low dispersion mirrors. The current work to raise the threshold is based on QWOT mirrors. Therefore, how to raise the threshold without sacrificing bandwidth and dispersion is

very meaningful. Several researchers have reported experimental results for LDMs [17]. Such studies on a large number of samples are known [16,18]. Several methods are recognized to be effective in increasing the threshold of QWOT [19]. The first method is to increase the damage resistance by using wide-bandgap-material protective layers. Depositing a highly damaged resistant layer stack on a conventional quarter-wave mirror (composite low dispersion mirrors, CLDM) overcomes the tradeoff between LIDT and bandwidth. Takada et al. proposed a broadband mirror with high damage fluence for chirped-pulsed amplification of 10 fs pulses. This mirror consists of broadband $TiO_2/SiO_2$ coatings and optimized high-damage-threshold $ZrO_2/SiO_2$ coatings [7]. Patel et al. demonstrated that a modified $Ta_2O_5/SiO_2$ structure in which the high index layer is replaced by either $HfO_2$ or $Y_2O_3$ for the three uppermost high-index layers results in an increase of ~50% in LIDT when measured at 1 μm using 350 ps laser pulses [15]. The second method to increase the damage threshold is electric field regulation. According to previous research, the damage caused to optical elements occurs at a laser power that yields a critical electric field intensity in the coating material [20]. Apfel et al. derived a mirror structure with thinner high-index layers, reducing the peak electric field in the high-index material by 33% and shifting the peak electric field away from the interface [21]. Bellum et al. used a similar method, where at an angle of incidence of 45° both the threshold of s- and p-polarization increased. [13]. Schiltz et al. observed that a modified-electric-field composite low dispersion mirror (E-CLDM) does not offer any definite damage performance advantages over the standard CLDM when tested for a 4 ns source and a 0.19 ns source [11]. However, the reason was not given in his report.

　　Compared with QWOT, the CLDM can increase the threshold without sacrificing the bandwidth by using wide-bandgap-material protective layers, and the method of electric field regulation can shift the peak electric field to the low-index material. Furthermore, the damage threshold in the femtosecond regime is more dependent on the electric field [12]. Therefore, we believe that combining the two methods can further increase the threshold in the femtosecond regime.

## 2. Materials, Design and Methods

　　We aim to design high-damage-threshold mirrors that meet a minimum reflectivity of 99.5% at a 45° incidence angle for p-polarization with GDD < ±300 fs$^2$ over a spectral range of 750–850 nm. We choose $SiO_2$ as a low-index material (~1.46), which is well-known for exhibiting high resistance to femtosecond laser damage. And $Ta_2O_5$ and $Al_2O_3$ is as a high-index material. In this study, we choose $Al_2O_3$ as the top high-index material which has a wide bandgap of 6.5 eV. The layer stack is represented by the formula G | (HL)$^m$ (XL)$^{n-1}$X | Air, where m is the periods of $Ta_2O_5/SiO_2$ stack; $n$ is the periods of $Al_2O_3$ layer, and H, L, and X denote $Ta_2O_5$, $SiO_2$, and $Al_2O_3$, respectively, G is substrate.

　　Figure 1a shows the calculated electric field distribution of CLDM for different protective layers. As the number of $Al_2O_3$ periods increases, the electric field in the narrow-bandgap material decreases. Meanwhile, the peak electric field intensity in $Al_2O_3$ remains constant. To ensure that the protective layer work, we need as many protective layers ($Al_2O_3$) as possible. However, different periods of the $Al_2O_3$ protective layer significantly impact the GDD performance, as shown in Figure 2a. The calculated GDDs of CLDM and E-CLDM are obtained by the film design software TFCalc. As the number of $Al_2O_3$ periods increases, GDD increases. When the number of periods $n$ is 5, the maximum GDD ripple is just under ±300 fs$^2$. Considering the electric field and dispersion, we replace $Ta_2O_5$ with $Al_2O_3$ in the five uppermost high-index layers. The layer stack is represented by the formula G | HL)$^{16}$(XL)$^4$X | Air. The initial design is a $Ta_2O_5$-$Al_2O_3/SiO_2$ composite mirror with 41 layers. This design combines the superior optical properties of $Ta_2O_5$ and the higher damage threshold of $Al_2O_3$. The layer structure of the quarter-wave optical-thickness composite mirror is shown in Figure 3a and the electric field intensity distribution in the proximity of the $Ta_2O_5$-$Al_2O_3$ interface is shown in Figure 1a ($n = 5$). The electric field distribution shows that the peaks' electric fields are located at the material interfaces,

which can easily lead to damage, and that the maximum field intensities at the $Al_2O_3$ and $Ta_2O_5$ interfaces are 1.2 and 0.33, respectively.

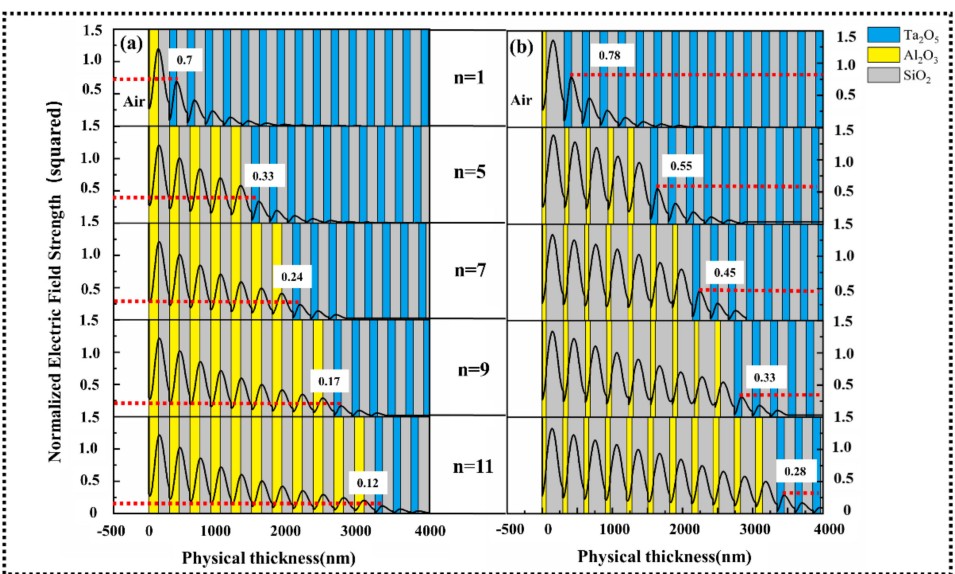

**Figure 1.** Calculated electric field distribution for different numbers of protective layer periods. (**a**) The electric field of a composite low-dispersion mirror (CLDM) for different numbers of protective layer periods. As the number of $Al_2O_3$ periods increases, the electric field in the narrow-bandgap material decreases. Meanwhile, the peak electric field intensity in $Al_2O_3$ remains constant. (**b**) The electric field of E-CLDM for different numbers of protective layer periods.

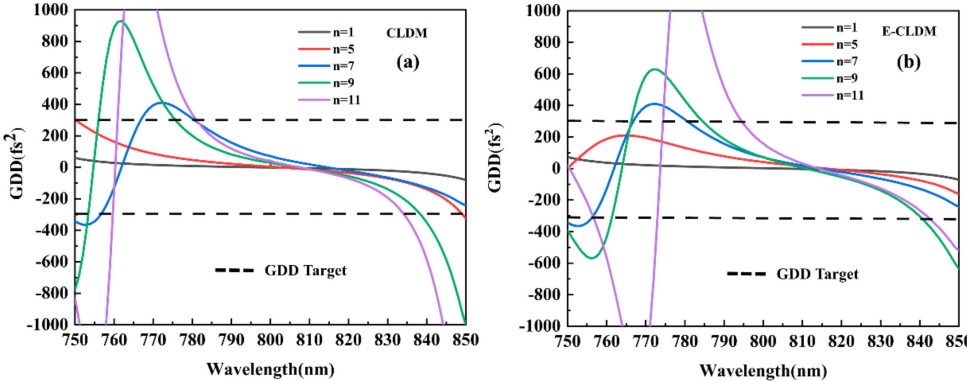

**Figure 2.** Dispersion for different numbers of protective layer periods. (**a**) CLDM, when $n = 5$, the maximum group delay dispersion (GDD) ripple is just under $\pm 300$ fs$^2$. When $n > 5$, the GDD ripple does not meet the requirements; (**b**) E-CLDM, when $n = 5$, the modified design has a superior dispersion value of $\pm 175$ fs$^2$, compared to the value of $\pm 300$ fs$^2$ for the CLDM stack. However, for $n > 5$, the GDD ripple does not meet the requirements.

The design E-CLDM, based on the initial design, is optimized such that the electric field is shifted away from the material interfaces and into the low-index high-damage threshold material ($SiO_2$). The electric field distribution of E-CLDM for different $n$ is shown in Figure 1b. We keep the electric field in $Al_2O_3$ and $SiO_2$ constant for different $n$. As the number of $Al_2O_3$ periods increases, the electric field in the $Ta_2O_5$ decreases. However, compared with the electric field in CLDM, the electric field in $Ta_2O_5$ of E-CLDM is higher. The layer profile of E-CLDM ($n = 5$) is shown in Figure 3b. It is obvious that the thickness of the $Al_2O_3$ is decreased, while that of the low-index material is increased [21]. The electric field distribution is shown in Figure 1b ($n = 5$), where the electric field intensity peaks are located in the low-index material and the maximum field intensities in $Al_2O_3$, $SiO_2$ and $Ta_2O_5$ are 0.8 and 0.55, respectively. At the same time, the maximum electric field intensity in $SiO_2$ increases to 1.35.

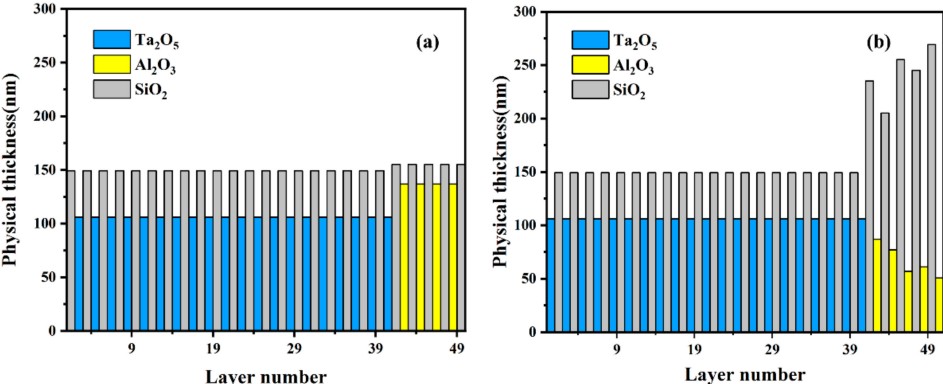

**Figure 3.** (**a**,**b**) Layer thickness profile of a CLDM stack of $Ta_2O_5$-$Al_2O_3$/$SiO_2$ (**a**) and that of a layer stack of $Ta_2O_5$-$Al_2O_3$/$SiO_2$ with reduced electric field intensity inside the $Al_2O_3$ layers.

Figure 2a,b shows a comparison between the theoretical GDD of the CLDM and E-CLDM stacks. For $n = 5$, the dispersion curve of E-CLDM exhibits a protrusion over the spectral range 750–760 nm, but still meets the design requirements in the working band (750–850 nm). The modified design has superior dispersion value of $\pm175$ fs$^2$, compared to the value of $\pm300$ fs$^2$ for the CLDM stack. However, for $n > 5$, neither CLDM nor E-CLDM meets the requirements of dispersion.

## 3. Fabrication

Mirrors based on Figure 3a,b were fabricated by dual-ion-beam sputtering technique (Spector, Veeco) with two ion sources (16 cm main source and 12 cm assistant source). The high-energy argon ions produced by 16 cm main source bombard the target (Ta, Al, $SiO_2$). The main function of the 12 cm assistant ion source is the pre-cleaning of the substrate and the improvement the coating absorption, making the coating more dense and bonding with the substrate stronger. The background pressure is $10^{-4}$ Pa and the substrate baking temperature is 120 °C.

## 4. Results

### 4.1. Reflectivity

The reflection spectrum was measured with a Perkin–Elmer (Lambda 1050) spectrophotometer. The reflectivities of the two sample types for p-polarization at 45° AOI are shown in Figure 4. This is consistent with that expected for the E-CLDM stack, which performs similarly to the CLDM stack. Both E-CLDM and CLDM stacks provide a reflectivity of 99.5% over a spectral range of 750–850 nm.

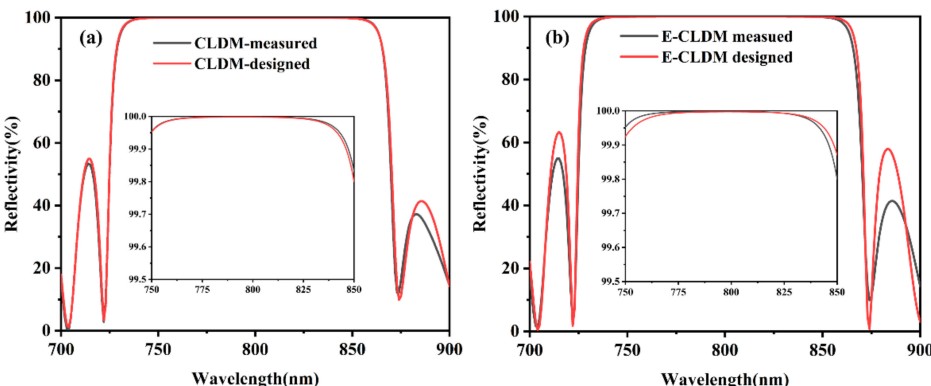

**Figure 4.** Measured and designed reflectivity of CLDM (**a**) and modified-electric-field CLDM (E-CLDM) (**b**) for p-polarization of 45° AOI.

### 4.2. Damage Threshold

A laser damage test for two types of samples was performed in the 1-on-1 mode [22]. Pulses were generated by a Ti:sapphire laser system with an 800 nm central wavelength and a pulse duration of 30 fs, and were p-polarized and incident at 45°. The spectral range is 800 ($\pm$35 nm). The measurement spot area is 0.221 mm$^2$ and the error range is 3%. The experimental setup is shown in Figure 5. A CCD camera was used to observe the damage in real time.

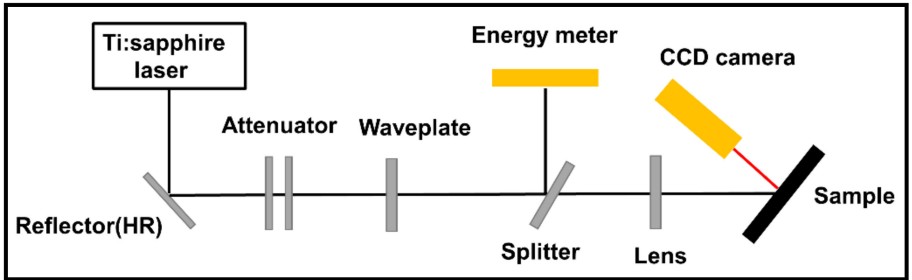

**Figure 5.** Experimental setup used for laser damage test.

Firstly, we measured the damage threshold of three low-dispersion mirrors, including CLDM, E-CLDM and 41 layers quarter-wave optical thickness mirror (Ta$_2$O$_5$/SiO$_2$), as shown in Figure 6. Compared to QWOT, which comprises a Ta$_2$O$_5$/SiO$_2$ stack and has a low damage threshold of 0.32 J/cm$^2$, the damage threshold was increased by 50% to 0.48 J/cm$^2$ for CLDM. However, contrary to our expectations, the damage threshold of E-CLDM was found to be lower than that of CLDM at only 0.41 J/cm$^2$. In other words, electric field regulation did not increase the damage threshold of the composite mirror. However, the damage thresholds of both CLDM and E-CLDM were higher than that of QWOT.

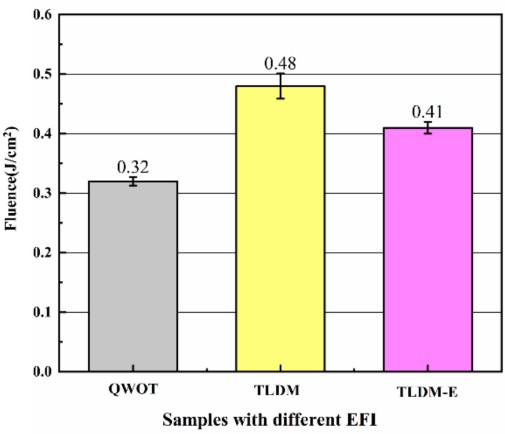

**Figure 6.** Damage threshold results for quarter-wave optical thickness (QWOT), CLDM, and E-CLDM. The thresholds of QWOT, CLDM and E-CLDM are 0.32, 0.48 and 0.41 J/cm$^2$, respectively. EFI—electric field intensity; Fluence—radiant energy per unit area.

### 4.3. Damage Morphology

To explain why electric field regulation would decrease the damage threshold of a composite mirror, the damage morphology of the E-CLDM stack was observed by scanning electron microscopy. As shown in Figure 7, the SEM results suggest that while the initial damage morphology (Figure 7a) occurred at a laser pulse fluence of 0.43 J/cm$^2$, the other two damage morphologies (Figure 7c,d) occurred at higher fluences. The beam is incident obliquely from the right side of the picture. Firstly, at a laser fluence of 0.43 J/cm$^2$, an inner layer material was melted and gasified, and the resulting gas pressure deformed the outer

layer, forming a protrusion. At a fluence of 0.45 J/cm$^2$, the internal pressure exceeded the limit stress of the layer, and thus, damaged the protrusion structure. As the fluence was further increased to 0.5 J/cm$^2$, the bulge structure was detached and exposed to a damage pit.

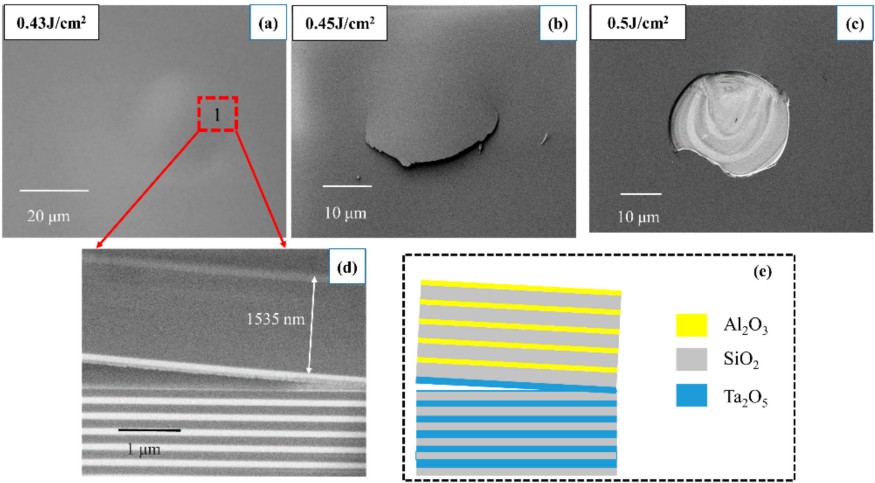

**Figure 7.** (**a**–**c**) surface damage morphology of E-CLDM at different fluences; (**d**) cross-sectional morphology of region 1 of the protrusion shown in (**a**); (**e**) diagrammatic sketch of (**d**). At a laser fluence of 0.43 J/cm$^2$, an inner layer material was melted and gasified, and the resulting gas pressure deformed the outer layer, forming a protrusion. At a fluence of 0.45 J/cm$^2$, the internal pressure exceeded the limit stress of the layer, and thus, damaged the protrusion structure. As the fluence was further increased to 0.5 J/cm$^2$, the bulge structure was detached and exposed to a damage pit. The real thickness of Al$_2$O$_3$/SiO$_2$ stack is consistent with the design thickness. The initial damage layer is located in the Ta$_2$O$_5$ layer.

To demonstrate the initial damage layer, we further observed the cross-sectional morphology after focused-ion-beam sectioning. The cross-sectional morphology of region 1 of the protrusion is shown in Figure 7d and the diagrammatic sketch of Figure 7d is shown in Figure 7e. Because the atomic number of Al is close to that of Si, the Al$_2$O$_3$/SiO$_2$ stack layer cannot be clearly observed. However, the first Al$_2$O$_3$ layer and Ta$_2$O$_5$ layer can be observed. A precise measurement shows that the distance between the first Al$_2$O$_3$ layer and the Ta$_2$O$_5$ layer is 1535 nm, and the design distance is 1542 nm. These data suggest that the position of the layer is accurate. The white layers are Ta$_2$O$_5$ and black layers are SiO$_2$.

It is unexpected that the initial damage site is located in the Ta$_2$O$_5$ layer. Taken together, these results suggest that the wide-bandgap material Al$_2$O$_3$ has no protective effect on the narrow-bandgap material Ta$_2$O$_5$. This causes the damage threshold of E-CLDM to be lower than that of CLDM.

## 5. Discussion

We next consider why the initial damage layer is Ta$_2$O$_5$. The threshold values presented in Section 4.2 were calculated without considering the electric field distribution. If the distribution is considered, internal thresholds that are characteristic for the layers will be obtained. Therefore, if the initial damage layer for different coatings is made of the same material, we can calculate the relations among damage thresholds of different designs from the electric field distribution. Thus, the theoretical damage threshold can be calculated according to Equation (1) [16,23].

$$F_t = F_{int} \times \left| E_{inc}^2 \right| / \left| E_{max}^2 \right| \tag{1}$$

Here $F_t$ is the theoretical damage threshold of the front surface of the sample, $F_{int}$ is the internal threshold of layer $X$, $E_{inc}$ is the magnitude of the incident electric field, and $E_{max}$ is the magnitude of the electric field inside layer $X$.

The internal threshold of a $Ta_2O_5$ layer can be calculated from the damage threshold of $Ta_2O_5/SiO_2$, as shown in Figure 6. The damage threshold of $SiO_2$ and $Al_2O_3$ layers can be obtained from the previous work of our group [24,25]. Thus, the internal thresholds of $Ta_2O_5$, $Al_2O_3$, and $SiO_2$ layers are 0.101, 0.337, and 0.430 J/cm$^2$, respectively. Figure 8 shows theoretical damage thresholds of CLDM or E-CLDM with different numbers of $Al_2O_3$ periods. The threshold of CLDM or E-CLDM is decided by the lowest threshold value in layer materials.

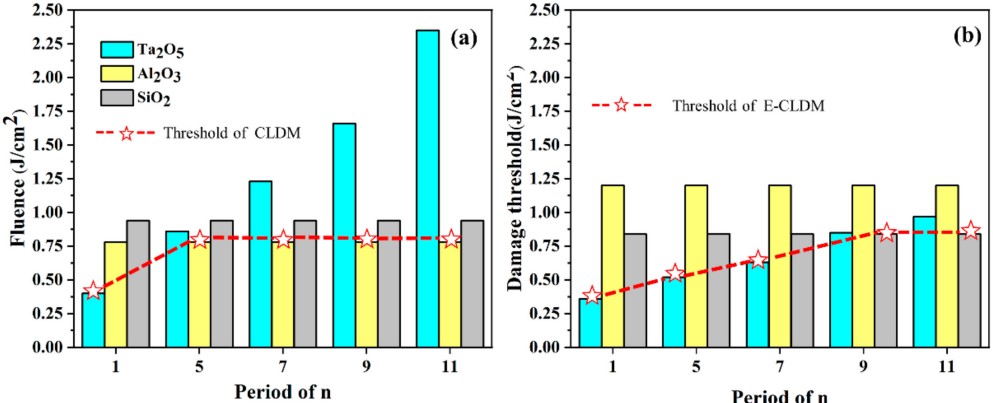

**Figure 8.** (**a**) Theoretical damage thresholds for all layer materials of CLDM with different protective layer periods. The threshold in $Ta_2O_5$ increases with the number of $Al_2O_3$ periods and the actual threshold of CLDM has the lowest threshold value for layer materials. $n = 5$ is an inflection point. $n < 5$, the damage threshold of E-CLDM increases with the $n$ increasing; $n > 5$, the threshold of CLDM remains unchanged. (**b**) Theoretical damage thresholds for all layer materials of E-CLDM with different protective layer periods. $n = 9$ is an inflection point. $n < 9$, the damage source is in $Ta_2O_5$. $n > 9$, the damage source is in $SiO_2$ and the threshold of E-CLDM keep constant.

From Figure 8a, a competitive relationship exists between $Ta_2O_5$ and $SiO_2$ in CLDM. We find that the threshold in $Ta_2O_5$ layer increases with the number of $Al_2O_3$ layer periods, because at different $Al_2O_3$ periods, the electric field in $Ta_2O_5$ decreases as shown in Figure 1b. So, if $n < 5$, the threshold in $Ta_2O_5$ layer is always the lowest value, that is, the damage source is in internal layer $Ta_2O_5$. In the case of $n = 5$ or $n > 5$, the electric field in $Ta_2O_5$ is at a very low level resulting in the threshold in $Ta_2O_5$ exceeds that in $Al_2O_3$. The damage source is in surface $Al_2O_3$ layer. The electric field distribution of E-CLDM for different $n$ is shown in Figure 1b. Compared with the electric field intensity of $Ta_2O_5$ in CLDM, the stack of E-CLDM has a higher intensity in $Ta_2O_5$. It implies that the surface electric field regulation causes the enhancement of the electric field in narrow band material $Ta_2O_5$. The threshold of E-CLDM with different numbers of $Al_2O_3$ periods is shown in Figure 8b. Compared with Figure 8a, the competitive relationship exists between $Ta_2O_5$ and $SiO_2$. If $n < 9$, the threshold of E-CLDM decreased because of the enhancement of the electric field in $Ta_2O_5$. $n = 9$ is an inflection point and the electric field in $Ta_2O_5$ is at a low level resulting in the threshold in $Ta_2O_5$ layer exceeds that of $SiO_2$. In the case of $n > 9$, the damage source is in $SiO_2$, and the threshold of E-CLDM keeps constant. So the optimal $n$ for E-CLDM is 9. The threshold of E-CLDM for $n > 9$ has no significant increase compared with CLDM. And considering the GDD, $n = 9$ is not a good choice. In summary, the method of shifting the electric field to wide-bandgap-material is no longer suitable for CLDM.

The competition effect can help us to obtain the best electric field control results for CLDM with different material combinations and different protective layer periods. We need to control the electric field according to the competition results of bandgap and the electric field of layer materials.

According to the electric field distribution shown in Figure 1 ($n = 5$), we calculate the theoretical threshold in $Ta_2O_5$, $Al_2O_3$, and $SiO_2$ layers of E-CLDM are 0.50, 1.20, and 0.84 J/cm$^2$, respectively. Clearly, the theoretical threshold of E-CLDM is 0.50 J/cm$^2$ and the initial damage layer is $Ta_2O_5$. The measured threshold is 0.41 J/cm$^2$. These data are consistent with the cross-sectional morphology, as shown in Figure 7d. Similarly, the theoretical thresholds in $Ta_2O_5$, $Al_2O_3$, and $SiO_2$ layer of CLDM for $n = 5$ are 0.86, 0.78, and 0.94 J/cm$^2$. The theoretical threshold of CLDM is 0.78 J/cm$^2$ and the measured threshold is 0.48 J/cm$^2$. Both the calculated damage thresholds are higher than the corresponding experimental thresholds. We attribute this discrepancy to low adhesion at the layer interfaces and residual stress that lower the threshold.

The internal threshold of $Al_2O_3$ is calculated through a single $Al_2O_3$ layer and that of $Ta_2O_5$ is calculated through $Ta_2O_5/SiO_2$. So this is the reason why there is a large deviation between the calculated result of CLDM and the experimental result. The relation between the calculated damage thresholds of the two samples (CLDM and E-CLDM) agrees with the experimental results.

## 6. Conclusions

In summary, we combined the methods of electric field regulation and wide-bandgap protective layers in the femtosecond regime. The damage threshold of two types of composite low-dispersion mirrors, with and without electric field regulation, are investigated using femtosecond laser pulses. The results show that the damage threshold of E-CLDM is lower than that of CLDM. The internal threshold is used to calculate the theoretical threshold of all samples. The damage threshold of CLDM or E-CLDM is determined by the competition results of bandgap and the electric field. The damage properties of layer materials are determined by bandgap and electric field distribution. The theoretical results show that the enhancement of the electric field in internal layers, caused by surface electric field regulation, is an immediate cause of the threshold to decrease. These data suggest that the method of shifting the electric field to wide-bandgap-material is no longer suitable for CLDM. The competitive effect between layer materials must be considered in the new electric field modulation method. The competition effect can help us to obtain the best electric field control results for CLDM with different material combinations and different protective layer periods. This study contributes to finding new ways to improve the LIDT of low-dispersion mirrors.

**Author Contributions:** Conceptualization: Y.W., J.S.; Data curation: Y.Z. (Yuhui Zhang), R.C.; Formal analysis: Y.Z. (Yuhui Zhang); Investigation, Y.Z. (Yuhui Zhang), Z.W.; Project administration: Y.W., H.H., D.L., Y.Z. (Yuanan Zhao), Y.J., K.Y., Y.S., Y.L., R.L. and J.S.; Validation: M.Z. and Y.J.; Writing—original draft: Y.Z. (Yuhui Zhang); Writing—review & editing: Y.W. All authors have read and agreed to the published version of the manuscript.

**Funding:** This work was supported by National Key R&D program of China (2018YFE0118000), Research on the Key Technology of TMT Large Aperture Wide Angle Spectral Dichroic Mirror (U1831211); the National Natural Science Foundation of China (Grant No. 11904376), NSAF Fund Jointly set up by the National Natural Science Foundation of China and the Chinese Academy of Engineering Physics (No. U1630140), the Youth Innovation Promotion Association, Chinese Academy of Sciences (CAS) (2017289), and the Strategic Priority Research Program of CAS (Grant No. XDB1603).

**Informed Consent Statement:** Informed consent was obtained from all subjects involved in the study.

**Data Availability Statement:** The data presented in this study are available on request from the corresponding author. The data are not publicly available due to privacy and secret.

**Conflicts of Interest:** The authors declare no conflict of interest.

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
