# Peer review of "Effect of Electric Field Regulation on Laser Damage of Composite Low-Dispersion Mirrors"

_coatings, doi:10.3390/coatings11010065_

Round 1

Reviewer 1 Report

The major criticism is related to the fact that the presented draft is written more like a technical report, not like a scientific paper.
What s the novelty in the presented research, what is different from the previous study?
What is the original contribution of the authors?
Technical comments
Could authors provide the simulation of the reflectivity of CLDM and E-CLDM stacks?
Results from figure 8 should be discussed more in detail, especially the difference between results presented in Figures a and b.
In the end, based on the criticism mentioned above, I am recommending the obligatory revision of the presented draft.

Author Response

Highlights

  • The method of electric field regulation was applied in composite low dispersion mirror(CLDM) in femtosecond laser damage regime.
  • The threshold of CLDM is decided by the results of competitive effect between layer materials.
  • The method of shifting the electric field to wide-bandgap-material is no longer suitable for CLDM. The competitive effect between layer materials must be considered in the new electric field modulation method.

The simulation of the reflectivity of CLDM and E-CLDM stacks the you mentioned is shown in Fig.4.

The difference between results presented in Figures (8) a and b is shown in line 218 to line 250.

Reviewer 2 Report

The authors demonstrated an interesting study where were combined the methods of electric field regulation and wide-bandgap protective layers in the femtosecond regime. Importantly there was noticed that CLDM made of various materials should be optimized according to the electric field competition effect. 

Some minor remarks should be addressed before publication:

  • the manuscript is not presented in a standard form, the section "Design and Fabrication" should be combined in Material and Methods, including sections e.g. chemicals, materials, fabrication, design, etc.
  • descriptions of the axis in figures are too small and of bad quality (especially fig.3)
  • if you write results and discussion it is as one section, in this case, I suggest separating "Results" from Discussion, cause anyway discussion is in a separate section.  

Author Response

Thank you very much for your valuable comments, we have made changes in the new manuscript according to your comments.

Reviewer 3 Report

In this manuscript, the authors reexamined two methods of field adjustment and wide bandgap protective layer fabrication in the femtosecond region to raise the laser-induced damage threshold for low-dispersion optics.

As a result, two results were drawn. 1)The CLDM or E-CLDM damage threshold is determined by the competitive outcome of the damage properties of the layered material.2) The increase in the electric field of the inner layer caused by the surface electric field adjustment is a direct cause of the lowering of the threshold.

However, I found that some major issues that need to be solved before the publication. Therefore, I do not suggest this manuscript to be published in Coating as it is.

Major issues

  1. L91: There is no explanation how to get the electric field distributions of CLDM and E-CLDM on Fig.1. Are they experimental results or calculated results?
  2. L131: There is no explanation how to get the GDDs of CLDM and E-CLDM on Fig.2.
  3. L217~L255: Theoretical calculations using cited references 16 and 22 alone are not convincing. I think the authors should actually show the experimental results using the mirror with the proposed structure.
  4. L286~L336: I think it's normal to write all author names in the cited papers or add “et al.” to the first author.
  5. L281~L283: In the section of “Acknowledgements”, if there is nothing to write, it should be omitted.

Reviewer 4 Report

This is an interesting manuscript, a useful contribution. My comments are shown below.

The authors discuss factors that influence the threshold for damage to low-dispersion mirrors from high-energy laser light. Two strategies have been used to increase the damage threshold. Composite dielectric low dispersion mirrors (CLDM) use upper layers of wide band gap material to protect the inner layers. E-CLDM mirrors are a modification that uses thinner protective layer. For thus study, the authors used low dispersion mirrors where the upper 5 layers of Ta2O5 were replaced with Al203. For both zones, alternate layers are composed of SiO2.

Research involved both experimental results and theoretical calculations. The authors concluded that the upper Al2O3 layers did not protect the underlying Ta2O5 layers from damage. Controlling electric field in upper layers resulted in electric field in internal layers. The authors conclude that the highest damage thresholds occur in CLDM mirrors, but the E-CLDM mirrors are superior to conventional quarter-wave optical thickness (QWOT) mirrors.

The manuscript is rich in analytical detail, and it is generally well-organized and clearly written. The content has both theoretical and practical value, and with minor revision I recommend it for acceptance.

My expertise in optics and electronics is limited, so my comments are somewhat general. I hope that another reviewer can better address technicalities.

Suggestions:

Abstract:

Line 20: The first sentence can be improved: “Low dispersion mirror has a promising application prospect….”. As written, the sentence refers to a single mirror, and says that this mirror has ownership of a prospect.  This phrasing appears to be an imperfect translation. I think a clearer statement would be something simple like “Low dispersion mirrors are important because of their potential use in petawatt (PW) laser systems”.

Line 29: Throughout the manuscript the undefined use of “competition” causes confusion. The wording of this sentence suggests that the competition is between band gap and electrical fields. Later in the manuscript competition appears to refer to the difference in properties between the upper protective layers and the underlying layers. Perhaps my ignorance of electronics is showing up here, but this an instance where improved clarity would make the paper more attractive to a wider range of readers.

A similar issue arises in regard to “fluence”. The legend for Figure 6 uses the abbreviation EFL without defining the term. Later, line 198 refers to “fluence values presented in section 3.2”, but section 3.2 does not use this term except as the EFL legend in Figure 6.  The manuscript uses many abbreviations for technical terms, but these terms are defined at their first use, which is an effective strategy. EFL is an exception. In the interest of broader readership, I suggest it would be helpful to explain “fluence” as being the radiant energy per unit area.

Section 3.2 shows the experimental setup where a CCD camera is used to observe damage that results from laser energy. The explanation is that “a CCD camera was used to observe the damage in real time. However, the manuscript does not report the results these observations. Instead, descriptions of damage to mirror layers comes from SEM images. The observations from the CCD camera need to be described.

The SEM images are excellent, but the descriptions of the damage features are not presented in the caption. That information is found in the text, but I believe clarity would be improved if those descriptions are instead included in the figure caption.

One other issue is that for the first time in the manuscript, the data include the QWOT (quarter wavelength optical mirror), in addition to the CLDM and E-CLDM mirror construction that are described in the introduction. Clarity would be improved by adding a mention of QWOT in the introduction.

The manuscript is well-organized, but the Introduction is rather unusual because it presents a summary of the results. This information would be appropriate for the abstract, but introductions are usually intended to present background information, not to summarize results.

The conclusions section provides a concise synopsis. The illustrations are excellent, and the reference list is thorough.

The remainder of my comments involve very small things:

Line 44. “Such studies on a large number of samples is known”. There is a plural/singular mismatch.  It should be “Such studies on a large number if samples are known.

Lines 68, 69. This is an example of a place where verb tenses are inconsistent, changing from past tense (“we combined….)  with present tense (“we replace….”). Similarly, line 84 uses the present tense “we aim…” and “we choose”. Whichever tense is chosen, it needs to be consistent.

Lines 77 and 263. Sentence begin with “And…”. This is an unfortunate sentence structure. In both cases the structure would be improved just be elimination “And”, while retaining the rest of the sentence.

Line 188.  “diagrammatic” does not need to be capitalized.

Line 261. The start “In a word,” is followed by a whole bunch of words. The phrasing would only be correct for an example like this: “In a word, no.”

Round 2

Reviewer 1 Report

I explained evryting in the letter yo the Editor

Reviewer 3 Report

The authors have rewritten the manuscript one after another according to the comments of the reviewers, and I think it is appropriate to publish it.